# ABTS/TAC Methodology: Main Milestones and Recent Applications

Antonio Cano , Ana B. Maestre , Josefa Hernández-Ruiz and Marino B. Arnao *

Department of Plant Biology (Plant Physiology), University of Murcia, 30100 Murcia, Spain
* Correspondence: marino@um.es

**Abstract:** ABTS (2,2′-azino-bis-(3-ethylbenzothiazoline-6-sulfonic) acid) is a widely used compound for determining the total antioxidant capacity (TAC) of plant extracts, food, clinical fluids, etc. This photometric assay is based on the reduction by the presence of antioxidant compounds of a well-known metastable radical ($ABTS^{\bullet+}$) which can be formed via several different approaches and be used in many different determination methodologies such as automated photometric measures in microplates, clinical robots, valuable titrations, and previous liquid chromatographic separation. Another interesting aspect is that, in some cases, the ABTS/TAC method permits sequential hydrophilic and lipophilic antioxidant activity determinations, obtaining total antioxidant activity values through the summary data of both types of antioxidants. In this work, we present a review of several aspects of the ABTS/TAC, highlighting the major achievements that have made this method so widely used, e.g., ABTS radical formation in hydrophilic or lipophilic reaction media, measurement strategies, automatization, and adaptation to high-throughput systems, as well as the pros and cons. Moreover, some recent examples of ABTS/TAC method applications in plant, human, and animal samples are discussed.

**Keywords:** ABTS assay; antioxidants; antioxidant activity; clinical samples; plant extracts; TAC; total antioxidant capacity





## 1. Introduction

One of the most common analyses carried out on plant extracts, food, clinical fluids, and other sources is that of antioxidant capacity. The 2,2′-azino-bis-(3-ethylbenzothiazoline-6-sulfonic acid, ABTS) method (ABTS/TAC or ABTS/TEAC) is one of the most widely used in different research areas such as food science technology, agriculture, plant science, and nutrition. Its widespread use in many aspects of research has resulted in an increasing in the numbers of citation over the years (Figure 1).

ABTS was initially used in the detection of fecal occult blood [1] and as a reagent in the determination of glucose [2] in a glucose oxidase/peroxidase assay. ABTS was also used in the determination of peroxidase activity and by kinetic studies [3–6]. The reaction product of ABTS is a radical cation of the oxidized form of ABTS ($ABTS^{\bullet+}$), which was later used in the determination of the antioxidant capacity of biological samples.

The ABTS/TAC assay is a spectrophotometric method that uses the oxidized ABTS radical cation ($ABTS^{\bullet+}$) to react with antioxidants to reduce the ABTS radical and lose its bluish green color. $ABTS^{\bullet+}$ has several characteristics that make it suitable for colorimetric assays; it has several absorbance peaks at different wavelengths [3,7], it has a high extinction coefficient, its solubility in water is high, and it is also soluble in organic media [8]. The redox potential ($E_0\prime$) for ABTS/$ABTS^{\bullet+}$ is 0.68 V [9], high enough to react with most antioxidant compounds [10,11]. There are two main mechanisms involved in the reaction between radicals and antioxidants, the hydrogen atom transfer reaction (HAT) involving a single-step movement of a hydrogen atom [12–14], and the electron transfer reaction

(SET) where a single electron is transferred to reduce a compound [13–15]. The ABTS assay mainly follows the SET mechanism, although HAT can also be applied [12].

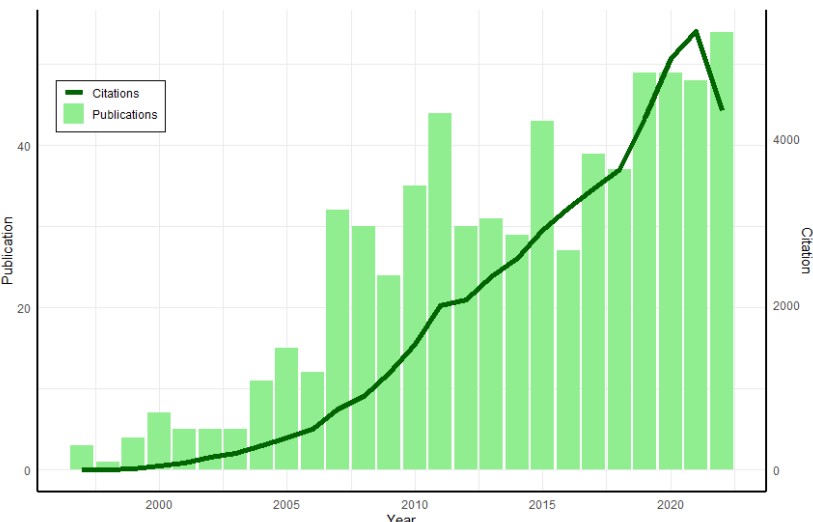

**Figure 1.** ABTS/TAC method publications and citations (1997–2022). Source: Web of Science, search: ABTS AND TEAC (all fields).

The first widely recognized use of $ABTS^{\bullet+}$ as a reactive to measure the antioxidant capacity of a sample or pure compound was by Miller et al. [16] using the reaction of myoglobin/$H_2O_2$ to produce the bluish green $ABTS^{\bullet+}$, although there were some previously published procedures from other authors (see below). Subsequently, the method was modified in certain aspects, such as the form of $ABTS^{\bullet+}$ generation and the parameter to estimate [7].

In this study, we review several aspects of the ABTS/TAC method, from ABTS radical generation to the different strategies used to determine photometric measurements. The major achievements that have led to this method being so widely applied, such as the possibilities of ABTS radical generation in hydrophilic or lipophilic reaction media, the possible automatization and adaptation to a high-throughput system, and several recent examples in different biological materials are discussed. Lastly, the pros and cons of these methods are analyzed.

## 2. Different Strategies for the ABTS/TAC Method

The generation of $ABTS^{\bullet+}$ can be achieved via different strategies, but enzymatic and chemical methods are the most widely used (Figure 2). Miller et al. [16] used the myoglobin/$H_2O_2$ system to oxidize ABTS; this procedure encloses many steps and is laborious to perform. Previously, the ABTS radical ($ABTS^{\bullet+}$) was generated using a $H_2O_2$/peroxidase (HRP) system for the determination of the flavonoid naringin [17] and indole-3-carbinol (indole-3-methanol) [18,19], but the method was not presented until 1996 by Arnao et al. [20]. Here, $ABTS^{\bullet+}$ was generated using a faster, easier, and a more controllable method, whereby the reaction medium is controlled by the $H_2O_2$ quantities present due to the well-known stoichiometry of the reaction (2 mol $ABTS^{\bullet+}$ per mol $H_2O_2$) [3,6]. Table 1 shows a chronological perspective of the ABTS/TAC method and its different approaches.

Subsequently, different chemical oxidative reagents have been used for the ABTS radical generation such as $MnO_2$ [21], potassium persulfate [22–24], $PbO_2$ [25], 2,2'-azobis-(2-amidopropane) (ABAP) [26], or $H_2O_2$ at low pH [27]. Other $ABTS^{\bullet+}$ generation methods include electrochemical oxidation [28,29] and peroxidase-like nanozyme [30,31].

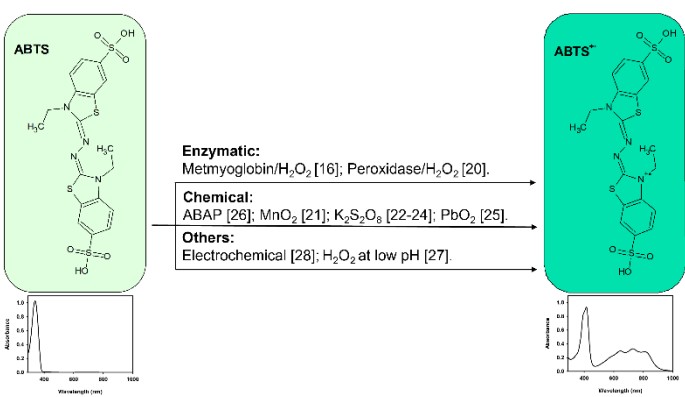

**Figure 2.** Generation strategies [16,20–25,27,28].

**Table 1.** Milestones in the development of the ABTS/TAC method.

| Year | Milestone | Ref. |
|---|---|---|
| 1990 | Horseradish peroxidase (HRP)/$H_2O_2$ generation method. First use of $ABTS^{\bullet+}$ for determining antioxidant activity of the flavonoid naringin. End-point strategy. | [17] |
| 1992 | HRP/$H_2O_2$ generation method. $ABTS^{\bullet+}$ used to estimate antioxidant activity of indole-3-carbinol. End-point strategy | [18,19] |
| 1993 | Myoglobin/$H_2O_2$ generation method. Kinetic strategy: reaction inhibition at fixed time. | [16] |
| 1996 | HRP/$H_2O_2$ generation method. Kinetic strategy: *lag* time. ABTS methods for carotenoid determination. | [20] |
| 1996 | $MnO_2$ oxidation method. End-point strategy. | [21] |
| 1999 | Microassay using microplate reader. HRP/$H_2O_2$ generation method. Kinetic strategy: reaction inhibition at fixed time. | [32] |
| 2000 | Direct generation of $ABTS^{\bullet+}$ in lipophilic media. HRP/$H_2O_2$ generation method. End-point strategy. Adaptation of lipophilic $ABTS^{\bullet+}$ to a microplate reader. | [8] |
| 2001 | Total antioxidant activity (TAA) as a combination of hydrophilic antioxidant activity (HAA) and lipophilic antioxidant activity (LAA) TAA = HAA + LAA. Use of $ABTS^{\bullet+}$ in both hydro- and lipophilic media. | [33] |
| 2001 | Adaptation of $ABTS^{\bullet+}$ to HPLC technique. | [34] |
| 2001 | Adaptation of $ABTS^{\bullet+}$ to stopped-flow technique. | [35] |
| 2002 | Electrochemical generation of $ABTS^{\bullet+}$. | [28] |
| 2003 | Adaptation of $ABTS^{\bullet+}$ to flow injection technique. | [36] |
| 2004 | $ABTS^{\bullet+}$ generation at low pH with $H_2O_2$. | [27] |
| 2019 | Paper-based device for $ABTS^{\bullet+}$ assay. | [37] |
| 2022 | Smartphone-based $ABTS^{\bullet+}$ assay, nanozyme use. | [30,31] |

$ABTS^{\bullet+}$ generation is the first step in this antioxidant assay, which can be achieved either in the presence of the antioxidant or before antioxidant addition. These different ways of generating the $ABTS^{\bullet+}$ lead to two different ways to quantify the antioxidant capacity: (i) kinetic and (ii) end-point approach [7]. In the ABTS method of Miller et al. [16], a kinetic approach was used, where the inhibition of the reaction due to the presence of the antioxidant is measured after a fixed time. Arnao et al. [17,20] also used this approach to determine the lag-time but recorded the delay in the appearance of the steady-state generation of $ABTS^{\bullet+}$. Thus, not only was the antioxidant capacity of the samples deter-

mined, but the possible inhibitory effects on the reaction by some compounds present in the samples were also evaluated [20].

In order to simplify the kinetic strategy measurements, several end-point methods were developed, in which media with preformed $ABTS^{\bullet+}$ were used [17–19]. In this case, the subsequent addition of the sample containing antioxidants produced a decoloration (bleaching) of the $ABTS^{\bullet+}$ [22,38,39]. This approach results in the method easiest to perform, eliminating the interaction of the antioxidant with the reactant enzymes, and making it suitable for high-throughput analysis. Although the end-point strategy has solved some of the problems observed in kinetic assays, new difficulties have arisen, such as the different reaction speeds of the different antioxidants present in the samples with $ABTS^{\bullet+}$, obtaining fast- and slow-reacting antioxidant types [40,41]. Recently, some novel and practical procedures to determine the antioxidant capacity have been proposed to improve ABTS/TAC determination through the estimation of exponential curve-fitting [24], inhibition percentage [27], or redox titration [25].

### 3. ABTS/TAC Methods for Hydrophilic and Lipophilic Antioxidants

Initially, the ABTS/TAC assay was developed for hydrophilic determination, estimating hydrophilic antioxidants such as organic acids, amino acids, glutathione, and phenols, among other antioxidants that can be solubilized in water. In this situation, fat-soluble antioxidants would apparently be dispersed in the aqueous medium, in a manner in which they could not be accurately measured. The development of a lipophilic method for determining these fat-soluble antioxidants was proposed by Miller [21] to measure the antioxidant capacity of carotenoids, using an ABTS radical generated in aqueous medium after dissolving carotenoid samples in hexane/acetone. Using the ability of horseradish peroxidase (HRP) to act in organic media, Cano et al. [8] proposed a modified ABTS/TAC method to generate $ABTS^{\bullet+}$ directly in organic media, where different organic solvents were tested for peroxidase reactivity and $ABTS^{\bullet+}$ stability.

The importance of obtaining a method which was capable of measuring both antioxidant capacities (hydrophilic and lipophilic) using the same reactive chromogen led to the possibility of the determination of the total antioxidant activity (TAA) as a combination of hydrophilic antioxidant activity (HAA) and lipophilic antioxidant activity (LAA) (TAA = HAA + LAA) (Figure 3). This was demonstrated in our studies in different plant material such as lettuce [42], tomato [43], grapes [44], citrus fruits [45,46], spinach [47], lupin [48], cereals [49], artichoke [50], *Quercus* tree [51], and chamomiles [52], as well as in foodstuffs such as vegetable soups [33], wine [53,54], and beers [38], and in animal material such as rat [55–57], canine [58], and human plasma [59], in addition to kidney, liver, and brain organs [56,60], and boar seminal samples [61]. Table 2 shows some of the obtained values using the ABTS/TAC method for different products, with a relative classification according to their antioxidant capacity.

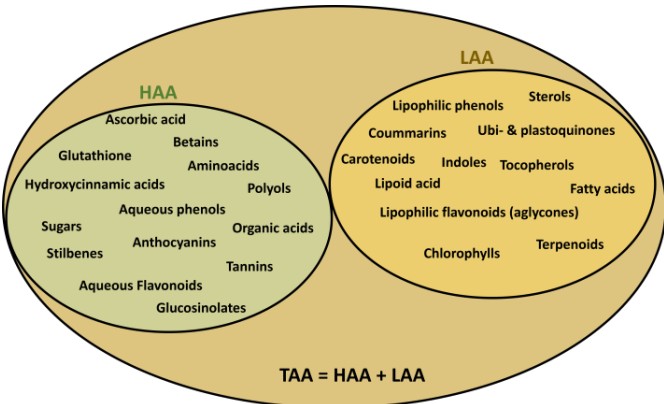

**Figure 3.** Total antioxidant activity (TAA) as summation of the hydrophilic (HAA) and lipophilic (LAA) activities, as well as their components.

**Table 2.** High, moderate, and low TAC ranking of different products according to published values.

| Sample | Type | TAC [a] | Ref. |
|---|---|---|---|
| Coffee | Beverage | 47.16 | [36,62] |
| Grape, black | Fruit | 35.93 | [36,44,63] |
| Blackberry | Fruit | 20.24 | [36] |
| Red wine | Beverage | 18.85 | [36,54,62,64] |
| Raspberry | Fruit | 16.79 | [36] |
| Artichoke | Vegetable | 15.03 | [36,50] |
| Black olive | Fruit | 14.73 | [36] |
| Redcurrant | Fruit | 14.05 | [36] |
| Blueberry | Fruit | 13.09 | [36] |
| Strawberry | Fruit | 11.05 | [36,63] |
| Pork | Meat | 3.50 | [65] |
| Grapefruit | Fruit | 3.43 | [36,45,63] |
| Radish | Vegetable | 3.22 | [36] |
| Orange juice | Beverage | 2.90 | [36,66,67] |
| Beef | Meat | 2.90 | [65] |
| Zucchini | Vegetable | 2.86 | [36] |
| Grapefruit juice | Beverage | 2.75 | [36,66,67] |
| Bean kidney | Vegetable | 2.70 | [63] |
| Lemon | Fruit | 2.68 | [45] |
| Chicken | Meat | 2.56 | [65] |
| Banana | Fruit | 0.64 | [36] |
| Flavored water | Beverage | 0.50 | [68] |
| Celery | Vegetable | 0.49 | [36] |
| Carrot | Vegetable | 0.44 | [36] |
| Cucumber | Vegetable | 0.43 | [36] |
| Fennel | Vegetable | 0.43 | [36] |
| Iceberg lettuce | Vegetable | 0.32 | [42] |
| Endive | Vegetable | 0.30 | [36] |
| Baby head lettuce | Vegetable | 0.24 | [42] |
| Soft drink | Beverage | 0.07 | [68] |

[a] TAC: total antioxidant capacity, expressed as mmol Trolox/kg FW (mean values).

## 4. Adaptation of the ABTS Assay to Different Techniques

The increase in the use of the ABTS/TAC assay for determining the antioxidant capacity and the rise in studies on the relationship between structure compound and antioxidant capacity led to the adaption of the different methods discussed for use in high-throughput systems. Adaptations have been made for several techniques such as microplate readers, high-performance liquid chromatography (HPLC), flow-injection assay (FIA), stopped-flow techniques, and automated analysis equipment (Figure 4).

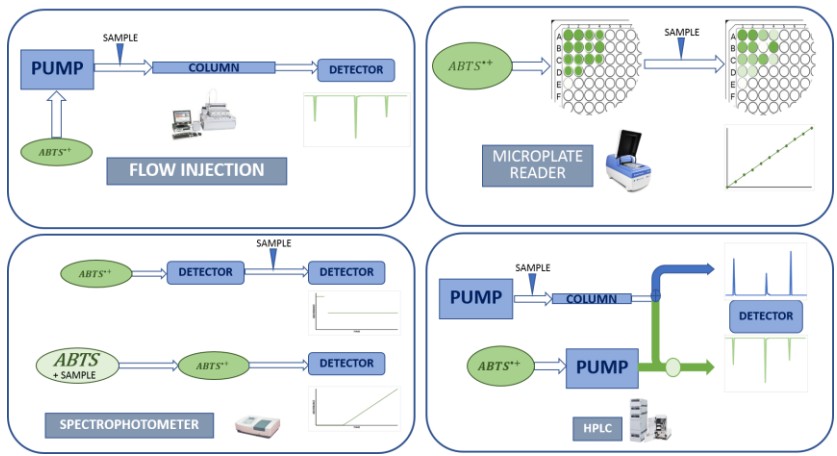

**Figure 4.** Adaptation of ABTS/TAC method to different high-throughput systems.

A microplate adaptation was developed by Laigth et al. in 1999 [32] for measuring antioxidant activity in rat plasma. The microplate assay was adapted to the preformed ABTS radical strategy [8,33] or in-well microplate ABTS radical generation [69]. Cano et al. [8] adapted the lipophilic ABTS/$H_2O_2$/HRP end-point method to a microplate reader for determining the antioxidant capacity of lipophilic α-tocopherol and β-carotene.

The ABTS/$H_2O_2$/HRP generation method was also adapted to an HPLC system, in a post-column ABTS radical reaction [70,71], and applied to both hydrophilic and lipophilic antioxidants. In this case, a better understanding of the peak separate compounds with antioxidant capacity was obtained, while quantifying the compound content and its respective hydro- or lipophilic antioxidant activity.

Recently, two new interesting adaptations have been developed, one using a paper device as support for the antioxidant determination using ABTS assay together with the Folin–Ciocâlteu and CUPRA assays [37], and the other using a peroxidase-like nanozyme to generate $ABTS^{\bullet+}$ [30], measuring the level of TAC with a portable device, while promoting a large-scale measurement of antioxidant activity using a smartphone [31].

## 5. Recent Applications of the ABTS/TAC Assay

Since the first use of this assay for determining the antioxidant capacity (see Table 1) it has become one of the most used methods, only surpassed by the DPPH method [45]. Many research fields have used the ABTS/TAC method in their studies, such as food technology, agriculture, plant science, nutrition, and clinical [7]. Some of the latest studies in different fields of research are discussed below.

In a study carried out in dogs, different ABTS/TAC methods including the method of Miller et al. with metmyoglobin [16], the method of $ABTS^{\bullet+}$ generated at low pH [27], and the method of Arnao et al. with peroxidase [38] were compared in terms of total antioxidant capacity determinations in the canine serum of healthy and inflammatory bowel disease-affected dogs [58]. The three ABTS/TAC methods assayed showed acceptable results, with an imprecision of less than 15%, but only the $H_2O_2$/peroxidase assay showed significant differences between the two dog samples, along with a higher correlation between observed and expected antioxidant activities in the canine sample dilution assay. After validation of the methods for their use in canine serum, the authors recommended metmyoglobin and peroxidase methods for ABTS radical generation [58].

A study on gestational diabetes mellitus in rats used the ABTS/TAC assay of Arnao et al. [38] for determining the hydrophilic antioxidant capacity in serum samples, describing a direct correlation between treatment and an increase in HAA, confirmed by a low oxidative marker [57,59].

The ABTS/TAC assay was used in the characterization and comparison of wild chamomile plants [52], using the hydrophilic and lipophilic determination of the peroxidase method [8]. In this study, differences between different studied chamomile plants in root, stem, leaf, and flower extracts were described, with relevant differences in HAA and LAA depending on the organ and the chamomile species [52]. Furthermore, an excellent correlation between the antioxidant capacities (HAA, LAA, and TAA) and the phenol and flavonoid content (aqueous, organic, and total) was found.

In another interesting study, the antioxidant activity of six byproducts from artichoke industrial processing was analyzed and compared [50]. These biowastes are very rich in phytochemicals with potential health benefits to humans and animals [72]. The content of phenols and flavonoids, and the antioxidant capacities, amongst other parameters, were analyzed, with selective thermal treatment in artichoke hearts and bracts; the most promising byproducts were found closest to the artichoke heart. In this type of study, the ABTS/TAC method was demonstrated to be a fast and reliable technique, in terms of the processing line, the antioxidant properties of food, and its byproducts.

In the last few years, some reviews on antioxidant activity assays have been published, where some controversies and limitations of different antioxidant assay approaches were discussed [7,13,73,74]. Certain problems in the use of persulfate for ABTS oxidation, and

difficulties in the kinetic approach for antioxidants with multiple hydroxyl groups have been described [13]. Apak et al. provided a list of desirable characteristics for an ideal antioxidant assay [13,14], most of which are accomplished by the ABTS/TAC assay using peroxidase. These characteristics include physiological pH (the ABTS assay can be run at different pH), stable and reproducible probes (the ABTS radical is a metastable radical with a peroxidative controlled reaction), activity against aqueous and organic antioxidants (the ABTS assay is compatible with both aqueous (HAA) and lipophilic (LAA) antioxidants), absorption in the visible spectrophotometric region, preferably beyond 500–700 nm to avoid interference from chlorophylls and anthocyanins ($ABTS^{\bullet+}$ has various spectral maxima at 414, 734, and 800 nm in the visible region), and optimal redox potential ($ABTS^{\bullet+}$ is able to oxidize the most important antioxidants) [13,14].

The important role of antioxidants in the response to oxidative stress in biological material and, therefore, in the prevention of many diseases and dysfunctions has increased the search for new antioxidant compounds, as well as the evaluation of their antioxidant activity in vitro. Obviously, these assays must be contrasted with other in vivo antioxidant studies. For example, some studies comparing the in vitro and in vivo antioxidant activities between different antioxidant assays were recently presented, highlighting the priority to determine the in vivo effect of antioxidants and its correlation with the in vitro antioxidant activity in interesting nutrients [75].

## 6. Conclusions

Despite some disadvantages observed using the ABTS/TAC method, which are shared with the commonly used DPPH method (not a natural physiological radical, large size, and slow vs. fast antioxidant reaction) [45], this method is still useful. The ABTS/TAC method is fast, with minimal processing, it is extremely versatile against different pH, different wavelengths can be selected to avoid spectrophotometric interferences, it can be used both for hydrophilic and lipophilic measurement, and it is easily adaptable to high-throughput methods (microplate, HPLC, etc.), thereby satisfying almost all requirements for an ideal antioxidant assay.

**Author Contributions:** A.C., J.H.-R., and A.B.M. contributed to the planning of the main ideas and visualization; A.C. was responsible first draft manuscript; A.B.M., A.C., and M.B.A. were responsible for the revision and final version. All authors have read and agreed to the published version of the manuscript.

**Funding:** This work was funded through the project of the Ministry of Science and Innovation "R+D+I Projects", the State Program for the Generation of Knowledge and Scientific and Technological Strengthening of the R+D+I System and R+D+I Oriented to the Challenges of Society of the State Plan for Scientific and Technical Research and Innovation 2017–2020, Grant PID2020-113029RB-I00 funded by MCIN/AEI/10.13039/501100011033. More information can be found at https://www.um.es/en/web/phytohormones/ (accessed on 30 December 2022; Phytohormones and Plant Development Lab).

**Data Availability Statement:** Not applicable.

**Conflicts of Interest:** The authors declare no conflict of interest.

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
