# Peer review of "ABTS/TAC Methodology: Main Milestones and Recent Applications"

_processes, doi:10.3390/pr11010185_

Round 1
Reviewer 1 Report
The Processes manuscript-2139527 deals with the ABTS/TAC methodology and focuses on its principle, adaptations and applications. In general, the document is well organized. First, different strategies for the ABTS/TAC method were described. Then, the chronological development of the technique was mentioned. The adaptability of the technique was discussed, however, most of the examples corresponded to works of the authors. It is suggested to include results from other authors to contrast the pros and cons mentioned in the document.
Minor corrections
177 line, "A recent study " phrase should be deleted since 1999 year is not considered as a recent one.
I consider that the processes-2139527 manuscript can be published in Processes, but the authors must include articles outside of their published work.
Author Response
REV#1
The Processes manuscript-2139527 deals with the ABTS/TAC methodology and focuses on its principle, adaptations and applications. In general, the document is well organized. First, different strategies for the ABTS/TAC method were described. Then, the chronological development of the technique was mentioned. The adaptability of the technique was discussed, however, most of the examples corresponded to works of the authors. It is suggested to include results from other authors to contrast the pros and cons mentioned in the document.
Minor corrections
177 line, "A recent study " phrase should be deleted since 1999 year is not considered as a recent one.
R: The bug has been fixed.
I consider that the processes-2139527 manuscript can be published in Processes, but the authors must include articles outside of their published work.
R: Thank you for your comments. In the new version (R1) we have included a new table (Table 2) with data of other authors who support this methodology.
Reviewer 2 Report
processes-2139527-peer-review-v1
Authors present a review of several aspects of the ABTS/TAC, highlighting the major achievements that have made this method so widely used, from ABTS radical formation in hydrophilic or lipophilic reaction media, measurement strategies, automatization and adaptation to high-throughput system, and pros and cons. Also, some recent examples of ABTS/TAC method applications in plant, human and animal samples have been commented.
The work is solid, excellent, it explains in a simple way everything related to this essay, for which in the future it will be a reference paper for researchers in the antioxidant theme.
Two suggestions are given below, after which the manuscript should be accepted for publication.
1- Considering the experience of the authors of the manuscript with the ABTS trial, if possible, at end of section 3 could be included a additional Table with the title that the authors consider appropriate.
In the Additional Table, the values obtained (in the units that the authors consider appropriate) in the different investigations of the authors with the ABTS assay should be included.
The values of the ABTS essay of the publications that the authors mention in the lines 125-130, page 4 , could be and others that the authors consider appropriate or important.
“This has been demonstrated in our studies in differ-125 ent plant material such as lettuce [42], tomato [43], grapes [44], citrus fruits [45,46], spinach 126 [47], lupin [48], cereals [49], artichoke [50], quercus tree [51], and chamomiles [52]; also in 127 foodstuffs as vegetable soups [33], wine [53,54], and beers [38]; and in animal material 128 such as rat [55–57], canine [58] and human plasma [59], and in kidney, liver and brain 129 organs [56,60], and also in boar seminal samples [61].”
The values of extracts, decoctions, of plants should be included with a ranking from highest to lowest antioxidant, this would be very useful to potential readers of the manuscript, it would give them a solid reference when considering or labeling their results in the ABTS assay. .
Please see page 1046 in A. Floegel et al. Journal of Food Composition and Analysis 24 (2011) 1043–1048 doi:10.1016/j.jfca.2011.01.008 for Table and a ranking suggested for the values obtained in the ABTS assay
2- If possible or appropriate, and considering their solid experience with the ABTS assay, the authors could suggest a range for the values obtained in the ABTS assay, which allows consideration of an extract, decoction, oil or other type of sample.
For what ABTS assay values, and in what units, would it be suggested that a medicinal plant extract, fruit, or other sample is ??:
Highly antioxidant
moderately antioxidant
slightly antioxidant
Inactive or devoid of antioxidant activity
There are reports of other biological activities, for example antimicrobial, where ranges of activity are suggested.
Please see reference on suggested MIC ranges to consider an extract or compounds with antimicrobial potential
For compounds, this stringent endpoints criteria were: significant (MIC<10μg/ml), moderate (10<MIC≤100 μg/ml) and low or negligible (MIC > 100 μg/ml) Kuete V, Efferth T (2010) Cameroonian medicinal plants: pharmacology and derived natural products. Front Pharmacol 1:123”
Also please see for antimicrobial activity
“Regarding edible plant extracts or their parts, it is estimated that they are very active if they show MIC values < 100 μg/mL, significantly active if 100 ≤ MIC ≤ 512 μg/mL, moderately active if 512 < MIC ≤ 2048 μg/mL and not very active if MIC > 2048 μg/mL (Tamokou et al., 2017).
Medicinal Spices and Vegetables from Africa
Therapeutic Potential Against Metabolic, Inflammatory, Infectious and Systemic Diseases2017, Pages 207-237
Medicinal Spices and Vegetables from Africa
Chapter 8 - Antimicrobial Activities of African Medicinal Spices and Vegetables
J.D.D.Tamokou, A.T.Mbaveng, V.Kuete,Tamokou, J.D.D., Mbaveng, A.T., Kuete.
Author Response
REV#2
processes-2139527-peer-review-v1
Authors present a review of several aspects of the ABTS/TAC, highlighting the major achievements that have made this method so widely used, from ABTS radical formation in hydrophilic or lipophilic reaction media, measurement strategies, automatization and adaptation to high-throughput system, and pros and cons. Also, some recent examples of ABTS/TAC method applications in plant, human and animal samples have been commented.
The work is solid, excellent, it explains in a simple way everything related to this essay, for which in the future it will be a reference paper for researchers in the antioxidant theme.
Two suggestions are given below, after which the manuscript should be accepted for publication.
1- Considering the experience of the authors of the manuscript with the ABTS trial, if possible, at end of section 3 could be included an additional Table with the title that the authors consider appropriate. In the Additional Table, the values obtained (in the units that the authors consider appropriate) in the different investigations of the authors with the ABTS assay should be included.
The values of the ABTS essay of the publications that the authors mention in the lines 125-130, page 4 , could be and others that the authors consider appropriate or important.
“This has been demonstrated in our studies in differ-125 ent plant material such as lettuce [42], tomato [43], grapes [44], citrus fruits [45,46], spinach 126 [47], lupin [48], cereals [49], artichoke [50], quercus tree [51], and chamomiles [52]; also in 127 foodstuffs as vegetable soups [33], wine [53,54], and beers [38]; and in animal material 128 such as rat [55–57], canine [58] and human plasma [59], and in kidney, liver and brain 129 organs [56,60], and also in boar seminal samples [61].”
The values of extracts, decoctions, of plants should be included with a ranking from highest to lowest antioxidant, this would be very useful to potential readers of the manuscript, it would give them a solid reference when considering or labeling their results in the ABTS assay. .
Please see page 1046 in A. Floegel et al. Journal of Food Composition and Analysis 24 (2011) 1043–1048 doi:10.1016/j.jfca.2011.01.008 for Table and a ranking suggested for the values obtained in the ABTS assay
2- If possible or appropriate, and considering their solid experience with the ABTS assay, the authors could suggest a range for the values obtained in the ABTS assay, which allows consideration of an extract, decoction, oil or other type of sample.
For what ABTS assay values, and in what units, would it be suggested that a medicinal plant extract, fruit, or other sample is ??:
Highly antioxidant
moderately antioxidant
slightly antioxidant
Inactive or devoid of antioxidant activity
There are reports of other biological activities, for example antimicrobial, where ranges of activity are suggested.
Please see reference on suggested MIC ranges to consider an extract or compounds with antimicrobial potential
For compounds, this stringent endpoints criteria were: significant (MIC<10μg/ml), moderate (10<MIC≤100 μg/ml) and low or negligible (MIC > 100 μg/ml) Kuete V, Efferth T (2010) Cameroonian medicinal plants: pharmacology and derived natural products. Front Pharmacol 1:123”
Also please see for antimicrobial activity
“Regarding edible plant extracts or their parts, it is estimated that they are very active if they show MIC values < 100 μg/mL, significantly active if 100 ≤ MIC ≤ 512 μg/mL, moderately active if 512 < MIC ≤ 2048 μg/mL and not very active if MIC > 2048 μg/mL (Tamokou et al., 2017). Medicinal Spices and Vegetables from Africa. Therapeutic Potential Against Metabolic, Inflammatory, Infectious and Systemic Diseases2017, Pages 207-237. Medicinal Spices and Vegetables from Africa. Chapter 8 - Antimicrobial Activities of African Medicinal Spices and Vegetables. J.D.D.Tamokou, A.T.Mbaveng, V.Kuete,Tamokou, J.D.D., Mbaveng, A.T., Kuete.
R: Thank you very much for your comments and suggestions. We have considered the two suggestions made and have chosen to introduce a new table (Table 2) that includes both. On the one hand, the ABTS/TAC values of several products, mainly fruits and vegetables, and the antioxidant activity ranking that shows their classification as high, moderate, or low TAC products. In addition to our references, we have included data and references of other authors, as both referees have suggested.
Round 2
Reviewer 1 Report
I consider that the processes-2139527 manuscript can now be published in Processes.